# How statistical learning interacts with the socioeconomic environment to shape children's language development

**Leyla Eghbalzad**[1]*, **Joanne A. Deocampo**[1], **Christopher M. Conway**[1,2¤]*

**1** Neurolearn Laboratory, Department of Psychology, Georgia State University, Atlanta, Georgia, United States of America, **2** Neuroscience Institute, Georgia State University, Atlanta, Georgia, United States of America

¤ Current address: Center for Childhood Deafness, Language, and Learning, Boys Town National Research Hospital, Omaha, Nebraska, United States of America
* Leghbalzad1@gsu.edu (LE); christopher.conway@boystown.org (CMC)

## Abstract

Language is acquired in part through statistical learning abilities that encode environmental regularities. Language development is also heavily influenced by social environmental factors such as socioeconomic status. However, it is unknown to what extent statistical learning interacts with SES to affect language outcomes. We measured event-related potentials in 26 children aged 8–12 while they performed a visual statistical learning task. Regression analyses indicated that children's learning performance moderated the relationship between socioeconomic status and both syntactic and vocabulary language comprehension scores. For children demonstrating high learning, socioeconomic status had a weaker effect on language compared to children showing low learning. These results suggest that high statistical learning ability can provide a buffer against the disadvantages associated with being raised in a lower socioeconomic status household.

## Introduction

The seeming ease with which typically developing children learn to comprehend and produce language suggests the existence of universal biological learning mechanisms [1, 2]. However, language development also relies on external interactions with social and environmental contexts [3, 4]. A combination of these perspectives, therefore, is needed to fully characterize language development [5]. In particular, it is important to study language development in children by focusing on the interaction between intrinsic (e.g., cognitive) and extrinsic factors (e.g., the social/linguistic environment).

The brain's ability to detect and encode probabilistic patterns in the environment, and then use such knowledge to predict upcoming stimuli and events, is thought to be a crucial component of cognition [6–13]. This type of learning, referred to as statistical learning, can take place without conscious awareness [14] and may unfold simultaneously as in the visual domain (e.g. pictures; [15]) or sequentially as in the auditory domain (e.g. musical tones; [16]. Statistical learning is a crucial component of visual perception [17], music perception and production [18], and language processing in infants [19, 20], children [21, 22], and adults [23, 24]. Several

**Data Availability Statement:** All data are available from the Open Science Framework database (https://osf.io/8cm2v/).

**Funding:** This research was funded by National Institute on Deafness and other Communication Disorders, Grant number: R01DC012037 (URL: https://www.nidcd.nih.gov). The funders had no role in study design, data collection and analysis, decision to publish, or preparation of the manuscript.

**Competing interests:** The authors have declared that no competing interests exist.

studies have linked such learning abilities to various aspects of language including both syntax [21, 25] and vocabulary [20].

In the classic statistical learning study conducted by Saffran, Aslin, and Newport, after only 2 minutes of exposure, infants were able to segment words from an auditory artificial speech stream by learning the transitional probabilities between syllables, providing an "existence proof" that such abilities might be used to detect word boundaries in natural speech [19]. In a longitudinal study, Shafto and colleagues investigated the relationship between serial learning and verbal and nonverbal language comprehension in infants 8–13 months. Using infants' looking time as a measure of learning, they reported that the infants' learning was correlated with vocabulary comprehension at the time of testing and with their gesture comprehension 5 months later [20]. Furthermore, in a study investigating the relationship between children's (ages 6–8) performance on a non-linguistic visual statistical learning task and language ability, Kidd and Arciuli [21] reported that statistical learning predicted performance on grammatical tasks involving passive and object-relative clauses. These studies demonstrate the important role of statistical learning in children's language development.

However, very little if any research has investigated how statistical learning abilities interact with social environmental factors such as the quality of the environment children are raised in. One of the most important indicators of the quality of environment is socioeconomic status (SES). SES consists of many distinct, but interrelated components which can individually and/or cumulatively influence children's development. These components include, but are not limited to, parental education level [26–40], household income [27, 29, 33, 37, 39, 41–44], home environment [45, 46], neighborhood safety, maternal mental health [35, 44], school type [37, 38], and stress level [28, 33, 38].

Various studies have found a relationship between children's poverty level and academic outcomes mediated by behavioral and/or neural measures [35, 36, 47]. Children who live in low SES families are reported to have less exposure to cognitive and linguistic stimulation and experience more stress in their environment [38, 48]. Consequently, children with low SES experience challenges with language proficiency as early as 18 months and by 24 months of age there is a 6-month vocabulary gap between groups of high and low SES children [49]. Low parental education, as an indicator of SES, is reported to predict language difficulties in children [40]. Hupp and colleagues [31] found that twenty-month-old children with well-educated mothers demonstrated better language production skills compared to those whose mothers were not as well-educated. Furthermore, variations in language exposure (e.g. quantity and quality of caregiver's input) are associated with changes both in neural function [37] and neural structure [50] of left lateralized language networks. Thus, socioeconomic disparity leads to an impoverished language environment, which in turn has a negative impact on children's language development.

Although it is known that language development is highly dependent on both the environment in which the child is raised as well as his or her own learning abilities, it is not known to what extent these two factors might interact to impact language development. For instance, it is possible that having better intrinsic pattern learning abilities could help offset the deleterious effects of being raised in an impoverished social environment. Alternatively, it is possible that learning ability cannot compensate for such environmental limitations. Answering this question is important as it can bring clarity into what factors can offset negative social environmental conditions, which in turn could inform intervention approaches.

## Current study

The specific aim of this study, therefore, was to examine the relationship between SES, statistical learning, and language performance in typically developing children. There were two main

research questions for this study: (a) to what extent does SES affect statistical learning performance in children? and (b) does statistical learning ability moderate the relationship between SES and language development?

We measured statistical learning by using the event-related potential (ERP) technique while children were performing a computerized visual, non-linguistic statistical learning task. The learning task used here is somewhat simple compared to other tasks used in prior statistical learning studies but has the advantage of being adapted to ERP methodology and is usable across a range of ages [51]. The task involves learning probabilistic relations among sequentially presented stimuli, in which some stimuli predict a target stimulus to varying degrees. Children are told to press a button every time they see a specific "target" stimulus (the exact target stimulus is randomly determined for each participant). Children view a serially-presented stream of stimuli, with the target occurring occasionally, similar to classic "oddball" tasks. However, what elevates this task above the simple oddball task is that unknown to the children, the target is partly predictable based on the preceding stimulus. One specific stimulus (the "high predictor") is followed by the target 90% of the time. Another stimulus (the "low predictor") is followed by the target only 20% of the time. Preceding the low or high predictors are a series of "filler" stimuli. As children learn the predictor-target associations, their response time to the target should be quicker when the target follows the high predictor relative to the low predictor, a finding which has been borne out in our previous work [52, 53].

In addition to reaction times, the use of ERPs provides a direct measure of neural responsiveness that may give a more sensitive measure of learning–or at least an additional, converging measure–compared to behavioral measures alone (e.g., [54]). Using ERPs, Jost et al. [51] observed a late positive component (400–700 milliseconds) in posterior electrode sites that was greater for the high predictor stimuli relative to the low predictor stimuli and was observed only in the 2nd half of the task, once learning was expected to have occurred. In a follow up study with adults using this same paradigm, Singh et al. found that this late positivity effect was significantly correlated with participants' explicit awareness of the predictor-target contingencies as assessed through subjective reports following the experiment [53]. Finally, Singh et al. observed this same component in typically developing children but an atypical profile in children with developmental dyslexia. In sum, this late positivity likely reflects perceptual or attentional processes associated with making a prediction about the upcoming stimulus [54].

Based on these previous findings [51, 52, 55], in the current study we predicted that if children learned the probabilistic relationships between the two types of predictor stimuli and the target, there should be significant differences in their response times (RTs) to the targets as well as differences in the ERP amplitudes to the predictors based on whether a trial was high or low probability. Specifically, we expected that reaction times would be quicker to the target when it was preceded by the HP predictor compared to the LP predictor and that ERP amplitudes would be significantly larger for the HP compared to the LP predictor. However, these effects are expected only to be present (or to be larger in magnitude) in the second half of the task, as the previous studies revealed that learning only occurred once a sufficient number of trials have been experienced [51].

In regard to the first research question of this study (does SES affect statistical learning performance?), research suggests that low SES may be associated with lower scores on a variety of cognitive abilities including working memory, cognitive control, and language [56]. However, no previous research has examined the effect of SES on statistical learning and therefore it is possible that statistical learning as measured by RTs and ERPs (quicker reactions and higher amplitudes to HP predictor compared to LP) could remain robust (i.e., unaffected) in the face of environmental influences, a perspective that is common in the implicit learning literature (e.g., [57]). In regard to the second research question (does statistical learning performance

moderate the effect of SES on language development?), we predicted that statistical learning as measured by RTs and ERPs (quicker reactions and higher amplitudes to HP predictor compared to LP) would provide a "buffer" against the effects that low SES has on language development. This would be shown by an interaction between statistical learning performance and parental education such that participants with low SES would show better language performance if they demonstrate high statistical learning relative to those demonstrating low learning.

## Method

### Participants

We recruited 42 typically developing, monolingual English-speaking children aged 7–12 from the Atlanta metropolitan area. Initially, two participants were excluded, one due to computer failure and one due to a hearing impairment in one ear. In addition, 14 participants, consisting of mostly 7-year-olds, did not meet the EEG criteria and were excluded from further analyses (see section on *Electroencephalography (EEG) recording and processing*). Consequently, our final sample included 26 participants ages 8–12 (age mean = 10.12 years, *SD* = 1.48; 17 males). We chose this age range for four reasons: (a) there has been relatively less research on statistical learning and other forms of implicit learning in middle childhood, with the bulk of the research having been done on infancy through preschool and adolescence through adulthood; (b) there are major changes in the development of both statistical learning and language at the endpoints of this age-range (around 7–8 years and around 12 years), but statistical learning seems to show little change within this age range [54, 58, 59]; (c) after 12 years, statistical learning appears to have reached adult levels [52] and thus seems less relevant to our focus here on child development; and (d) the pattern learning task (with EEG measures) used in this study has been difficult to use with younger children, but has successfully been used with this age range [51, 55].

During their visit to our lab in the Psychology Department at Georgia State University, parents/guardians provided written informed consent and children assent to participate. The entire session lasted about 3 hours. Participants were offered a toy, worth $10 for participating and parents received $50. This study was approved by the Institutional Review Board (IRB) of Georgia State University and is in accordance with ethical standards described within the Declaration of Helsinki.

### Measures

The statistical learning task will be described in detail below. SES was measured through the use of a demographic questionnaire. To assess language, we included two measures of children's language development (receptive vocabulary and grammaticality judgment) using standardized assessments because both aspects of language have been reported to be related to statistical learning [20, 21, 25, 52]. Finally, we incorporated several measures of general cognitive ability to control for the effect of other cognitive abilities such as working memory and selective attention.

**Socioeconomic status (SES).** Parents completed a questionnaire providing socioeconomic status (SES) and demographic information. This questionnaire consisted of questions about their individual and household income, education, and other demographics of the primary and secondary caregivers and child. We used the average of both caregivers' education level as the measure of SES. Caregivers' education level was rated using the following scale: 0 = Less than High School; 1 = High School; 2 = Some college; 3 = Associate's degree; 4 = Bachelor's degree; 5 = Master's degree; 6 = PhD/Professional degree. Household income was not

**Table 1. Participant characteristics and descriptive statistics.**

|  | N = 26 | Low SES group | High SES group |
|---|---|---|---|
|  |  | N = 15 | N = 11 |
| **Age in years** | 10.12 ± 1.48[a] | 10.53 ± 1.45 | 9.55 ± 1.39 |
| **Gender** |  |  |  |
| Male | 17 | 10 | 7 |
| Female | 9 | 5 | 4 |
| **Race** |  |  |  |
| American Indian or Alaskan Native | 1 | 0 | 1 |
| lack or African American | 8 | 7 | 1 |
| White/Caucasian | 12 | 6 | 6 |
| More than one race | 5 | 2 | 3 |
| **Parents/Caregivers Education average** |  |  |  |
| 0. Did not graduate high school | 3 | 3 |  |
| 1. High school | 5 | 5 |  |
| 2. Some college | 1 | 1 |  |
| 3. Associate's degree | 2 | 2 |  |
| 4. Bachelor's degree | 4 | 4 |  |
| 5. Master's degree | 8 |  | 8 |
| 6. Ph.D./ Professional degree | 3 |  | 3 |
| **PPVT st** | 123.27 ± 21.33 | 107.8 ± 24.02 | 120.73 ± 14.93 |
| **Grammaticality Judgement st** | 104.15 ± 14.76 | 97.8 ± 14.76 | 112.82 ± 9.84 |
| **Block Design st** | 10.85 ± 3.89 | 10.13 ± 4.09 | 11.82 ± 3.55 |
| **Digit Span st** | 10.69 ± 3.53 | 10.73 ± 4.06 | 10.64 ± 2.84 |
| **Stroop st** | 50.69 ± 8.97 | 50.2 ± 10.07 | 51.36 ± 7.62 |
| **ERP H-L 1st half** | 2.22 ± 1.66 | 2.24 ± 1.61 | 2.19 ± 1.81 |
| **ERP H-L 2nd half** | 2.93 ± 2.78 | 2.95 ± 2.78 | 2.92 ± 2.91 |
| **RT L-H 1st half** | 40.03 ± 50.22 | 28.15 ± 43.98 | 54.08 ± 55.49 |
| **RT L-H 2nd half** | 88.1 ± 117.6 | 47.01 ± 93.98 | 136.66 ± 128.14 |

[a] mean ± SD.

st = Standard scores.

PPVT = Peabody Picture Vocabulary Task.

ERP = Event-related potential.

H-L = Difference in amplitude between High and Low probability conditions.

RT = Response time.

L-H = Difference in response time between Low and High probability conditions.

used in the analyses due to more than half of participants opting not to report income. A summary of average education levels for participants' caregivers is reported in Table 1.

**Statistical learning task.** The visual statistical learning task was based on a computer task developed by our research group [51] that in turn was based on the classic visual oddball paradigm, but with probabilistic regularities (transitional probabilities) embedded in the stimulus stream. We made the task child-friendly by making it into a game with a background story. Children were told a story about a magician who tried to make food for his children using his magic hat. Children were instructed to "catch" the food by pressing a button as quickly as possible.

The participants viewed a stream of flashing stimuli consisting of hats of different colors presented one at a time with a black background. Each stimulus was presented for 500

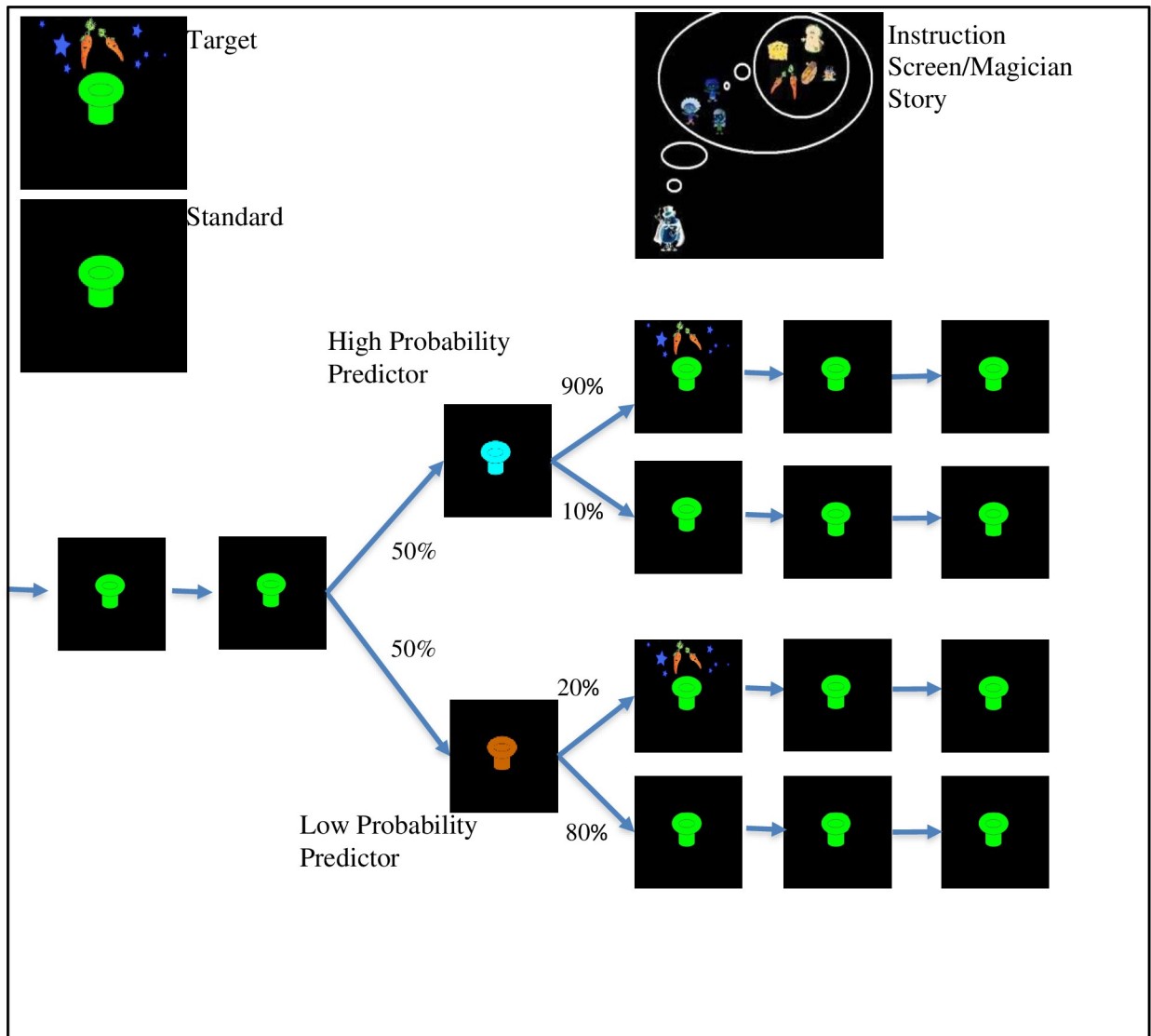

**Fig 1. Schematic representation of the statistical learning task.** The low predictor and high predictor stimuli were presented with equal frequency (i.e., the same number of trials); however, the target followed the high predictor on 90% of high predictor trials but only followed the low predictor on 20% of low predictor trials.

milliseconds and was followed by a black screen for 500 milliseconds. Occasionally, a target hat with food, to which children were instructed to press a button, was presented within the stream. Participants were not told that hats of different colors differentially predicted the probability of occurrence of the target hat (which was indicated by food presence above the hat, not hat color). Targets followed predictors with varying transitional probabilities (see Fig 1): the target hat followed the high predictor hat (HP) 90% of the time and the low predictor hat (LP) 20% of the time. In addition, targets occasionally followed "filler" (or standard) stimuli without a predictor. This was done to decrease the overall predictability of the target's occurrence and make the task more difficult.

Each experimental condition (HP and LP) contained 60 trials, and there were 60 filler trials as well (half with a target and half without). Six blocks of 30 trials each (10 trials per predictor

condition and 10 filler trials) were separated by 30-second breaks during which children watched a short cartoon related to the magician story. Task duration was roughly 20 minutes. Thus, although from the perspective of the participant the presentation of stimuli seemed to be a continuous stream of mostly one color of hat (the filler) occasionally punctuated by another color hat (one color for HP and one for LP) which might or might not be followed by the target (a hat of the same color as the filler with food above it), trials were constructed to include a random number of fillers from 1 to 7 followed by a high predictor or low predictor (or another filler if it was the occasional target appearing without a predictor). The high predictor was then followed by the target 90% of the time (or another filler 10% of the time) and the low predictor was followed by a filler 80% of the time (or the target 20% of the time). The trial ended with one more filler stimulus, however, to the participant, the stream just continued. Color of filler, HP, and LP were randomly assigned to each participant.

The entire statistical learning task consisted of a continuous learning task with no separate learning and testing phases as is often found in other statistical learning tasks. EEG was collected continuously throughout as children were incidentally learning the relationships between HP and the target and LP and the target. They were expected to show learning over time as demonstrated throughout the task without any explicit probe. Learning was operationally defined as differences in ERPs and response times between conditions (HP or LP) that developed over time, which was measured by examining the first half versus the second half of the task.

**Electroencephalography (EEG) recording and processing.** ERPs reflecting stimulus-time-locked changes in electrical potential on the scalp during the statistical learning task were collected using an Electrical Geodesic, Inc. 32-channel sensor net in a 132 square foot double-walled, sound-proofed acoustic chamber. Impedances were kept below 50 kΩ. Continuous EEG was acquired with a .1 to 100 Hz band-pass filter digitized at 250 Hz with a vertex reference, later re-referenced to the average reference and resampled at 256 Hz. ERPs were time-locked to the onset of each predictor stimulus (epochs: -200 ms to +900 ms; see below). This resulted in 60 trials for each of the two predictor conditions (HP and LP).

EEG data were pre-processed, including 0.1 Hz high-pass filtering, 30 Hz low-pass filtering, and baseline correction to the 200 ms pre-stimulus EEG, using Net Station Version 4.5.4 (Electrical Geodesics, Inc.). The remainder of processing was done using a combination of custom scripts and pre-programmed GUI functions in MATLAB (versions R2012b 8.0.0783, R2018a 9.4.0, and R2020a 9.8.0; MathWorks) [60] and the EEGLAB Toolbox (versions 10.2.2.24a and 2019.1 [61]) for MATLAB. Bad channel data were removed and then replaced using a spherical interpolation of data from surrounding channels. Independent component analysis (ICA) procedures were applied to the continuous EEG to find and remove eye blink and eye movement components. The EEG was subsequently epoched to a 1100 ms window from 200 ms before stimulus presentation to 900 ms after stimulus presentation. Other movements and artifacts were removed manually after epoching. Participants were required to have a minimum of 15 good epochs per condition in the first half of the SL task and 15 per condition during the second half of the SL task to be included in further analyses; all of the participants in this sample were well above this threshold. On average, 27.7% of epochs were rejected.

Data from 6 sensors in the posterior region of interest were extracted for analysis, an *a priori* decision based on the findings of Jost et al. [51]. Data from perimeter sensors (2 sensors) and the two sensors directly behind the isolated common sensor (COM) were not included in analyses due to electromyogram (EMG) noise and other excessive noise. All other posterior region sensors were included (see Fig 2). Data were grand averaged across trials, electrodes, and participants for each condition and each half of the acquisition session to produce ERP waveforms. Mean amplitudes were extracted for the 400–700 ms post-stimulus time window,

also an *a priori* decision based on the findings from Jost et al. [51]. To reiterate, region of inter-est (ROI) and time window for analysis were chosen *a priori* based on previous research in the literature as recommended by Keil et al., [62] and Luck and Gaspelin [63] in their guidelines for the publication of EEG studies. Based on the ERPs observed in the statistical learning task described by Jost et al. [51], we chose to analyze the ERP time window 400–700 ms post-stimu-lus presentation from the posterior ROI. This is the time window and ROI in which Jost et al. [51] and Singh et al. [55] found ERP effects of predictor condition in a similar statistical learn-ing paradigm and with children of similar ages.

**Language assessments.**   We used 2 standardized language assessments. The Peabody Pic-ture Vocabulary Test, Fourth Edition (PPVT-4) [64] assessed receptive vocabulary using pic-tures. The Grammaticality Judgment subtest of the Comprehensive Assessment of Spoken Language (CASL) [65] assessed grammar by orally presenting a sentence with or without a grammatical error, and the child was asked whether it sounded correct and if not to fix it by changing one word. Age-based standard scores were used to score both of these tests (see Table 1).

**Cognitive assessments.**   We used 3 cognitive assessments to measure participants' general cognitive ability: the Stroop Color and Word Test: Children's Version [66] as a measure of executive function and 2 subtests of the Wechsler Intelligence Scale for Children Fourth Edi-tion Integrated (WISC-IV Integrated) [67]: Block Design as a measure of spatial ability and Digit Span as a measure of verbal short term and working memory. Standard scores for inter-ference between color word reading and color naming were used in the Stroop Test. Standard scores were also used for Block Design and Digit Span (see Table 1).

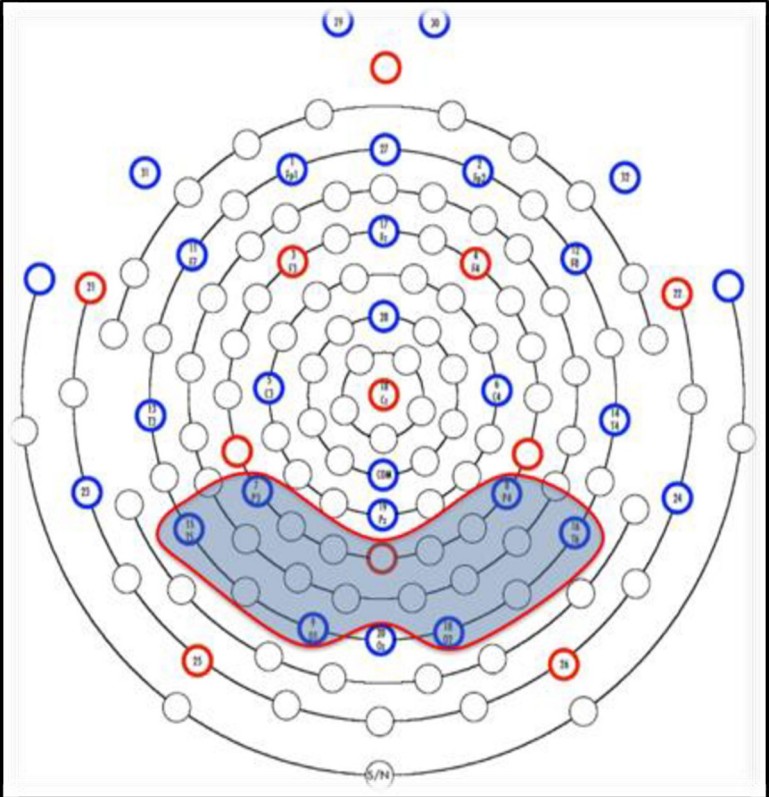

**Fig 2. Sensor map for the EEG 32-sensor net.** The region of interest is shaded in gray.

The tasks and assessments were administered in the same order for all participants: PPVT, visual learning task (about 20 minutes long), CASL, Stroop, Digit Span, and Block Design.

## Results

### Behavioral evidence of statistical learning

We used a two-way repeated-measures ANOVA with the factors of probability (high vs. low) and block (first half vs. second half) to determine whether there were differences in the behavioral response times for each of the 2 probability conditions across the first half (first 60 trials) and second half (last 60 trials) of the task. Due to technical issues, the response times for 2 participants were not recorded during the ERP data acquisition; however, the ERP responses for these participants were intact. Therefore, we excluded these participants only in analyses that include response time data ($N$ = 24; 15 males). The results showed that there was a significant main effect of predictor condition ($F(1,23)$ = 16.65, $p < .0001$); however, the main effect of block was not significant ($F(1,23)$ = 0.11, $p$ = ns). More importantly, there was a significant interaction between predictor condition and block, $F(1,23)$ = 6.14, $p < .05$. Results of post-hoc paired-sample t-tests (using adjusted Bonferroni alpha levels of 0.025 per test) showed that in the first half of the task participants responded significantly quicker to the targets in the high (M = 404.68, SD = 73.4) compared to the low-probability condition (M = 444.71, SD = 66.55), t(23) = -3.91, p = .001. The results for the second half of the task also revealed significantly quicker responses to the target for the high (M = 377.22, SD = 86.06) compared to the low-probability condition (M = 465.32, SD = 77.85,), t(23) = - 3.67, p = .001 which suggest that learning was evident in both halves of the task.

To further investigate the nature of the interaction and capture the magnitude of the learning effects in analyses, we created difference scores between high- and low-probability predictor conditions for the RT means (L-H). As mentioned above, the participants exhibited shorter response times for high- compared to low-probability conditions; therefore, we created L-H difference scores to avoid analyzing negative response times. Results of paired-sample t-tests showed a significant increase in the RT L-H difference score variable from the first (M = 40.03, SD = 50.22) to the second half (M = 88.10, SD = 117.60), $t(23)$ = - 2.48, $p$ = .021. These results show that the magnitude of learning is greater in the second half which provides behavioral evidence that prolonged exposure to the embedded probabilistic patterns increased children's learning of the underlying regularities. Fig 3 shows the change in participants' RT difference scores from the first half to the second half of the visual learning task.

### Neurophysiological evidence of statistical learning

Based on the results of Jost and colleagues [51], we focused our analyses on the posterior region of the scalp during the 400–700 ms post predictor time window, using a predefined set of 6 electrodes (see Fig 2). Fig 4 displays the grand averaged ERP waveforms in this region for the first half (Fig 4A) and second half (Fig 4B) of the task. To illustrate the distribution of EEG activity across the scalp, we created topographical 2-D maps by using EEGLAB toolbox (version 2019.1) [61] in MATLAB software (version R2012b 8.0.0783) [60]. Maps in Fig 5 show the ERP wave amplitude averaged across all 32 electrodes for each probability condition (high, low) and block (first vs. second half) in the 400–700 ms time-window.

A two-way repeated-measures ANOVA with the factors of probability condition (high, low) and block (first vs. second half) in the 400–700 ms time-window did not show a significant main effect of block ($F(1,25)$ = 2.0, $p$ = ns) or predictor condition ($F(1,25)$ = 3.75, $p$ = ns); however, there was a significant interaction between block and predictor condition, $F(1, 25)$ = 14.71, $p$ = 0.001. Post-hoc paired-samples t-tests (using adjusted Bonferroni alpha levels of

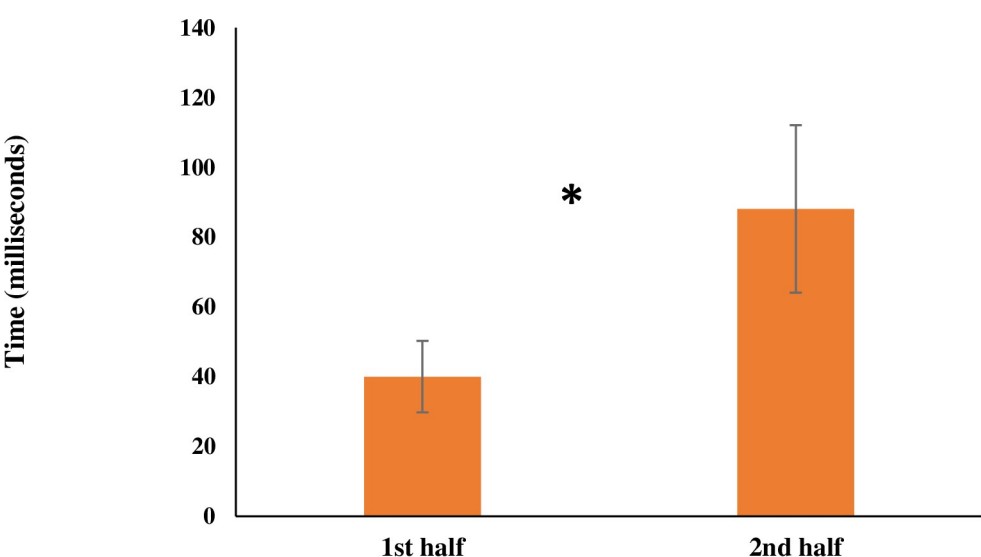

**Fig 3. Behavioral change in learning.** This figure shows change in RT (difference in response time between Low and High predictor conditions) from the first half to the second half of the statistical learning task. Error bars represent *standard error*.

0.025 per test) indicated that in the first half of the task, the ERP wave amplitudes were not significantly different between the high ($M$ = 2.14, $SD$ = 2.74) and low-probability conditions ($M$ = 2.55, $SD$ = 2.87), $t(25)$ = - 0.76, $p$ = 0.46. However, in the second half of the task, the amplitudes were significantly higher for the high ($M$ = 4.12, $SD$ = 3.94) compared to the low-probability condition ($M$ = 1.82, $SD$ = 3.07), $t(25)$ = 3.51, $p$ = .002.

Overall, the ERP results show that the children's ERPs demonstrated sensitivity to the different probability conditions only in the second half of the task, providing further evidence that children had learned the probabilistic relationships between predictor and target stimuli after this amount of exposure, similar to the findings by Jost and colleagues [51].

For the next set of analyses exploring the relationship among SES, statistical learning, and language, we created difference scores between high- and low-probability predictor conditions for the ERP amplitudes (H-L) to capture learning as a single variable. Due to statistically significant learning observed in the second half of the task, we focus these analyses on RT and ERP difference scores in the second half of the task only. In addition, absolute values of the ERP difference scores were used because conceptually, any difference in amplitudes between high and low predictor conditions is an indication that these conditions have been differentiated, i.e., that learning occurred. Exploratory analyses confirmed that the same general effects were observed whether or not absolute values were used, but appeared to be more robust using this approach.

## Correlation analyses

The relationship between statistical learning ability (measured using the RT and ERP amplitude difference scores in the second half of the task), parental education, and neuropsychological assessments (raw language and cognitive measures scores) were examined using a partial correlation analysis controlling for age of the participants. We found significant correlations

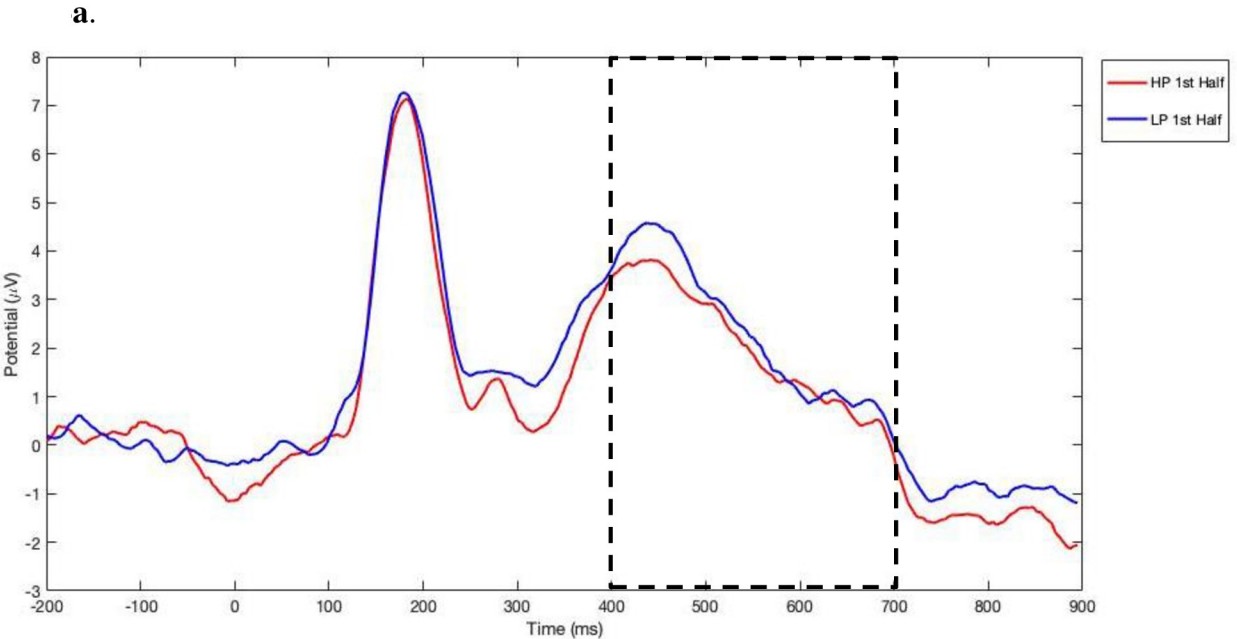

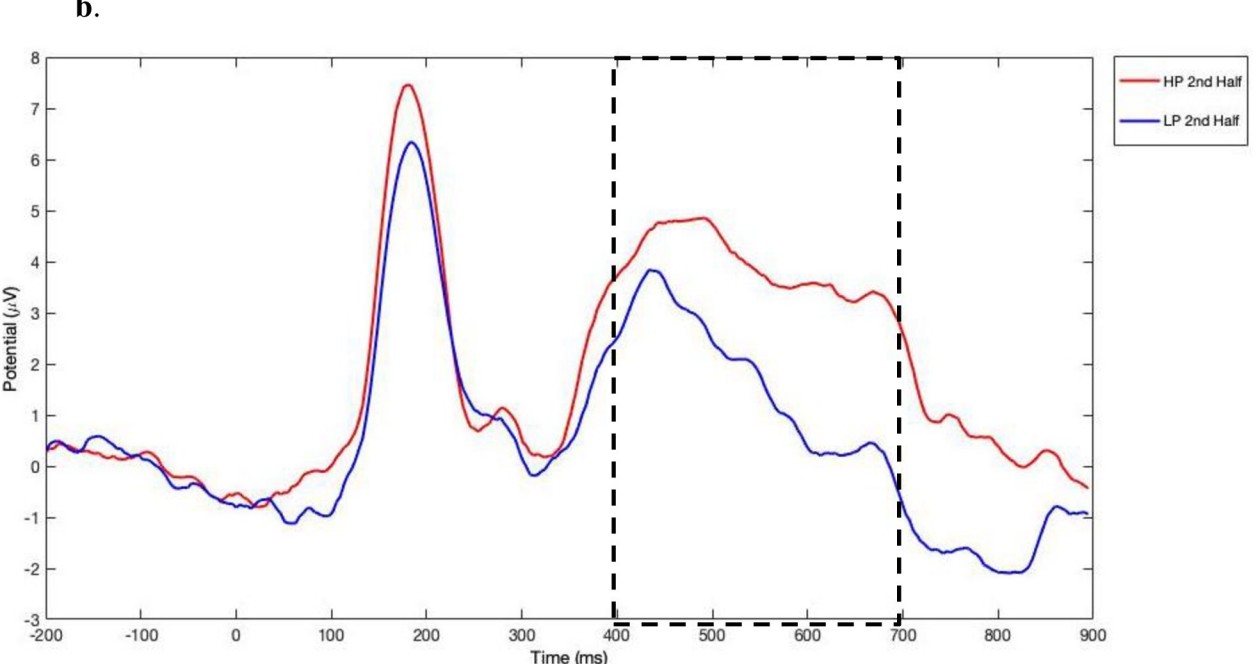

**Fig 4. ERP average waveforms in the posterior region elicited during the statistical learning task.** (a) average waveforms in the first half of the task. (b) average waveforms in the second half of the task. High-probability: red; low-probability: blue.

between parental education and performance on language assessments. Parental education ($M = 3.65$, $SD = 1.93$) was positively correlated with raw scores on both PPVT ($M = 168.81$, $SD = 32.45$), $r = 0.61$, $p = .002$, and Grammaticality Judgment ($M = 57.92$, $SD = 16.0$), $r = 0.60$,

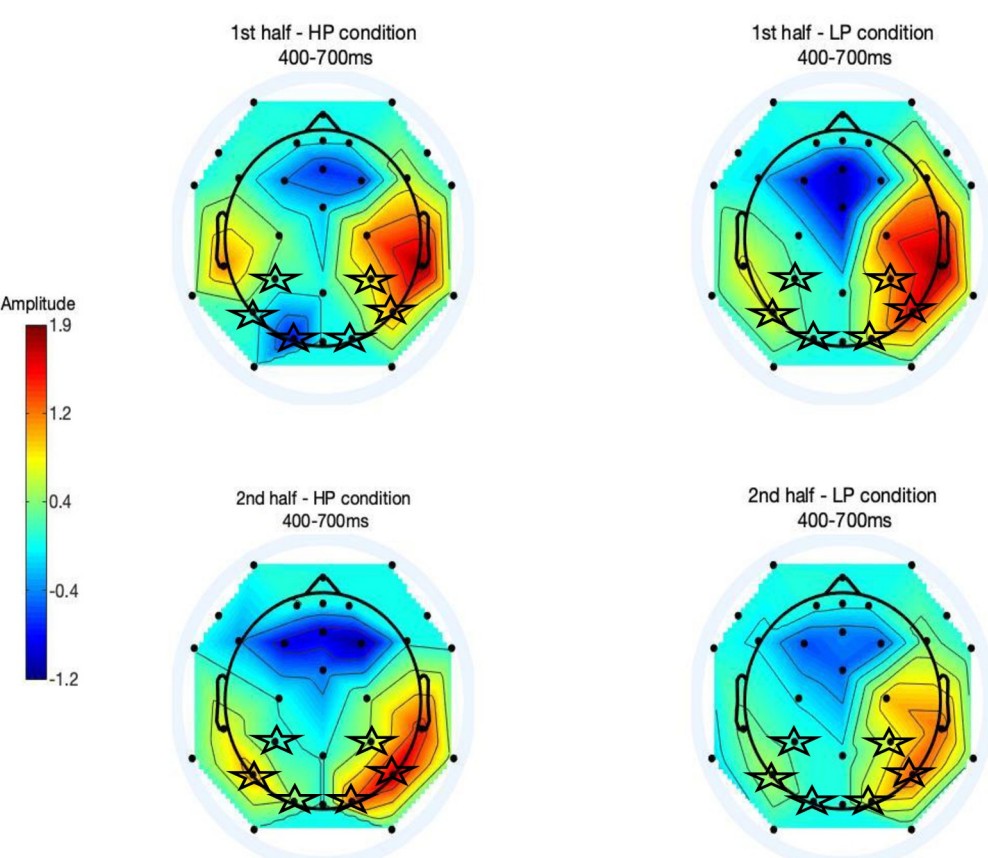

**Fig 5. Topographical maps depicting ERP amplitudes averaged across electrodes in 400–700 millisecond time-window after predictor onset.** Maps are shown for the first half of the task for the high-probability (top left) and the low-probability (top right) conditions. Likewise, maps are shown for the second half of the task for the high-probability (bottom left) and low-probability conditions (bottom right). The 6 electrodes of the *a priori* posterior region used in the waveform analyses are marked by stars.

$p$ = .002. These results suggest that higher parental education level is associated with children performing better on language assessments. On the other hand, neither of the statistical learning measures (RTs or ERPs) was significantly correlated with either of the language measures nor with parental education level. A complete list of correlation results and descriptive statistics of all measures are reported in Table 2.

## Influence of SES on statistical learning

To answer our first research question to examine the possible effect that SES might have on statistical learning, we used Pearson correlations to investigate a potential relation between the neurophysiological and behavioral measures of learning in the second half of the task and parental education level. As reported in Table 2, in our sample, parental education level did not directly correlate with neurophysiological or behavioral performance on the statistical learning task.

## Moderation analyses

To answer our second research question, we used hierarchical multiple regression models to specifically examine the moderating effect of statistical learning performance on the

**Table 2. Pearson correlations (controlling for age) and descriptive statistics.**

|  | 1 | 2 | 3 | 4 | 5 | 6 | 7 | 8 |
|---|---|---|---|---|---|---|---|---|
| **1. Parental education level** | - - - | | | | | | | |
| **2. Grammaticality Judgement** | **0.60**$^{**}$ | - - - | | | | | | |
| **3. PPVT** | **0.61**$^{**}$ | **0.89**$^{**}$ | - - - | | | | | |
| **4. Block Design** | 0.40 | **0.72**$^{**}$ | **0.63**$^{**}$ | - - - | | | | |
| **5. Digit Span** | 0.06 | **0.61**$^{**}$ | 0.51* | 0.49* | - - - | | | |
| **6. Stroop** | 0.06 | -0.39 | -0.12 | -0.38 | -0.17 | - - - | | |
| **7. ERP H-L 2nd half of the task** | 0.02 | -0.26 | -0.14 | -0.45* | -0.34 | 0.52* | - - - | |
| **8. RT L-H 2nd half of the task** | 0.31 | 0.52 | 0.22 | -0.06 | -0.08 | -0.12 | 0.11 | - - - |
| **Mean** | 3.65 | 57.92 | 168.81 | 36.23 | 16.31 | 50.69 | 2.93 | 88.10 |
| **SD** | 1.94 | 16.0 | 32.48 | 14.86 | 4.12 | 8.96 | 2.78 | 117.60 |

*$p<$ .05

**$p<$ .01

coefficients in **bold** remained significant after Bonferroni correction, p = 0.006.

PPVT = Peabody Picture Vocabulary Task.

ERP = event-related potential.

H-L = difference in amplitude between High and Low probability conditions.

RT = response time.

L-H = difference in response time between Low and High probability conditions.

relationship between SES and language. Before entering the variables in the hierarchical multiple regression, parental education and statistical learning scores (moderator) were standardized to reduce multicollinearity between the variables. We conducted separate regression analyses for RT and ERP amplitude measures, using PPVT and Grammaticality Judgment scores as outcome variables in each regression analysis (4 regression analyses total). As recommended by Frazier, Tix, and Barron [68], in the first step of each regression, we entered parental education and statistical learning (RT or ERP) as the predictor and moderator, respectively, and the language measure (performance on PPVT or Grammaticality Judgment test) was entered as the outcome variable. In the second step of the model, we entered the interaction term of parental education and statistical learning performance. Prior to conducting the hierarchical multiple regressions, relevant assumptions were tested. First, the assumptions of singularity and collinearity were met as predictor variables (parental education and statistical learning measures) were not combinations of each other and were not highly correlated (see Table 2). Additionally, collinearity statistics (Tolerance and VIF) were within acceptable limits [69, 70]. Second, the assumptions of normality, linearity, and homoscedasticity were met according to the scatterplots and histograms of standardized residuals of the data [70]. Because statistical learning was evident in the second half of the task, regression analyses were performed for behavioral (RT) and ERP data from the second half only. Results showed that there was no significant moderating effect of the behavioral measure of statistical learning on the relationship between parental education and either language measure (PPVT: $F(2,23) = 5.10$, $p = .23$; Grammaticality Judgment: $F(2,23) = 6.59$, $p = .16$).

However, results of the regression analysis using the neurophysiological measure showed that statistical learning performance as measured by the difference in ERP amplitudes between probability conditions was a significant moderator of the relationship between parental education and both language measures (PPVT & Grammaticality Judgment). With parental education and H-L ERP as predictor variables, the model significantly explained 38% of the variance in children's performance on the PPVT test, $R^2_{adj} = .382$, $F(2, 23) = 8.73$, $p = .002$. There was a

main effect of parental education level on PPVT performance, $\beta = .65$, $p < .001$, but not a main effect of statistical learning performance (H-L) $\beta = -.19$, $p =$ ns. Adding the interaction term to the model significantly increased the variance explained to 47%, $R^2_{adj} = .475$, $F(3, 22) = 8.24$, $p = .044$ (see Table 3).

Similarly, with parental education and H-L ERP as predictor variables, the model significantly explained 46% of the variance in children's performance on the Grammaticality Judgment test, $R^2_{adj} = .46$, $F(2, 23) = 11.76$, $p$ .001. There was a main effect of parental education level on Grammaticality Judgment performance, $\beta = 0.69$, $p < .001$, but not a main effect of statistical learning ability (H-L) $\beta = -0.25$, $p =$ ns. Adding the interaction term to the model significantly increased the variance explained to 55%, $R^2_{adj} = .55$, $F(3, 22) = 11.30$, $p = .027$ (see Table 4).

The significant interactions between parental education and statistical learning performance in the second half of the task are depicted in Fig 6A for PPVT scores and Fig 6B for Grammaticality Judgement scores. According to simple slope analysis following regressions, for children who showed high levels of statistical learning, parental education level did not influence either PPVT ($\beta = 0.32$, $t(24) = 1.48$, $p =$ ns) or grammaticality judgement ($\beta = 0.32$, $t(24) = 1.80$, $p =$ ns) scores. However, for children who showed low statistical learning, parental education had a significant influence on both PPVT ($\beta = 0.99$, $t(24) = 4.54$, $p < 0.001$) and grammaticality judgement scores ($\beta = 1.04$, $t(24) = 5.20$, $p < 0.001$).

As an additional control, we included each of the cognitive measures (Stroop task, Block design, & Digit span) as covariates in the moderation analyses with parental education and the statistical learning ERP H-L variable for the second half of the task. After adding the Stroop task to the analysis, the moderating effect of statistical learning performance on the relationship between parental education and performance on PPVT ($R^2_{adj} = .60$, $F(4, 21) = 10.35$, $p = .017$) and Grammaticality Judgment ($R^2_{adj} = .63$, $F(4, 21) = 11.85$, $p = .012$) remained significant. Adding Block Design and Digit Span changed the significant moderating effect of statistical learning performance on the relationship between parental education and performance on PPVT (Block design: $F(4, 21) = 10.10$, $p <$ ns; Digit Span: $F(4, 21) = 7.82$, $p <$ ns). However,

**Table 3. Hierarchical regression analysis of the moderating effect of statistical learning (ERPs) on the relationship between parental education average and PPVT scores.**

| | Unstandardized Coefficients | | Standardized Coefficients | t | Δ R2 |
|---|---|---|---|---|---|
| | b | SE | β | | |
| Step 1 | | | | | 0.38** |
| PPVT | 91.42 | 7.73 | | 11.83** | |
| Parent education | 7.12 | 1.74 | 0.65 | 4.10** | |
| ERP H-L 2nd half | -1.42 | 1.21 | -0.19 | -1.17 | |
| Step 2 | | | | | 0.46** |
| PPVT | 76.06 | 10.17 | | 7.48** | |
| Parent education | 11.14 | 2.48 | 1.01 | 4.49** | |
| ERP H-L 2nd half | 3.93 | 2.75 | 0.51 | 1.43 | |
| Parent education X ERP H-L 2nd half | -1.34 | 0.63 | -0.87 | -2.14* | |

$N = 26$.

$*p < .05$

$**p < .01$.

ERP = event-related potential.

H-L = difference in amplitude between High and Low probability conditions.

**Table 4. Hierarchical regression analysis of the moderating effect of statistical learning (ERPs) on the relationship between parental education average and grammaticality judgement scores.**

| | Unstandardized Coefficients | | Standardized Coefficients | t | Δ R2 |
|---|---|---|---|---|---|
| | b | SE | β | | |
| **Step 1** | | | | | 0.46** |
| Grammaticality Judgement | 88.85 | 4.99 | | 17.81** | |
| Parental education | 5.25 | 1.12 | 0.69 | 4.68** | |
| ERP H-L 2nd half | -1.32 | 0.78 | -0.25 | -1.69 | |
| **Step 2** | | | | | 0.55* |
| Grammaticality Judgement | 78.05 | 6.44 | | 12.13** | |
| Parental education | 8.08 | 1.57 | 1.06 | 5.14** | |
| ERP H-L 2nd half | 2.44 | 1.74 | 0.46 | 1.40 | |
| Parental education X ERP H-L 2nd half | -0.95 | 0.40 | -0.89 | -2.37* | |

*N* = 26.

$^*p < .05$

$^{**}p < .001$.

ERP = event-related potential.

H-L = difference in amplitude between High and Low probability conditions.

after adding Block Design and Digit Span the moderating effect of statistical learning on the relationship between parental education SES and performance on Grammaticality Judgment remained significant, $R^2_{adj}$ = .69, $F(4, 21)$ = 14.16, $p$ = .049 and $R^2_{adj}$ = .72, $F(4, 21)$ = 17.40, $p$ = .025, respectively. These results suggest that the moderating effect of statistical learning performance on the relationship between parental education and grammar ability is independent of these other cognitive factors.

## Discussion

In this study, we investigated the relationships among SES (measured by parental education level), visual statistical learning ability (operationally defined by the difference in ERP amplitudes and response times between High- and Low-probability conditions; [51, 52, 55]), and language outcomes in children. In the statistical learning task, children demonstrated sensitivity to the different probability conditions, as measured by both RTs and ERPs, indicating learning of the statistical probabilities. Statistical learning was especially pronounced in the second half of the task. Note that in this task, the HP and LP stimuli occur with equal frequency; thus, statistical learning is based on differences in the transitional probabilities between the HP and LP stimuli and the target, not based on simple frequency discrimination.

Consistent with previous findings [40, 46, 47] there was a positive relationship between children's SES level (as measured by caregiver education levels) and both syntactic knowledge (Grammaticality Judgment subtest of the CASL) and receptive vocabulary knowledge (PPVT). Children with more highly educated caregivers demonstrated better receptive vocabulary and grammar knowledge skills, compared to those children whose caregivers were not as highly educated. On the other hand, SES was not correlated with statistical learning ability, nor did SES group assignment (low or high) have a significant effect on statistical learning.

Although SES did not directly impact statistical learning, importantly, the results of the moderation analyses revealed that children displaying high levels of statistical learning (as measured by ERPs) appeared to have more robust syntactic language ability as well as higher levels of vocabulary development that was less affected by their SES. In other words, the

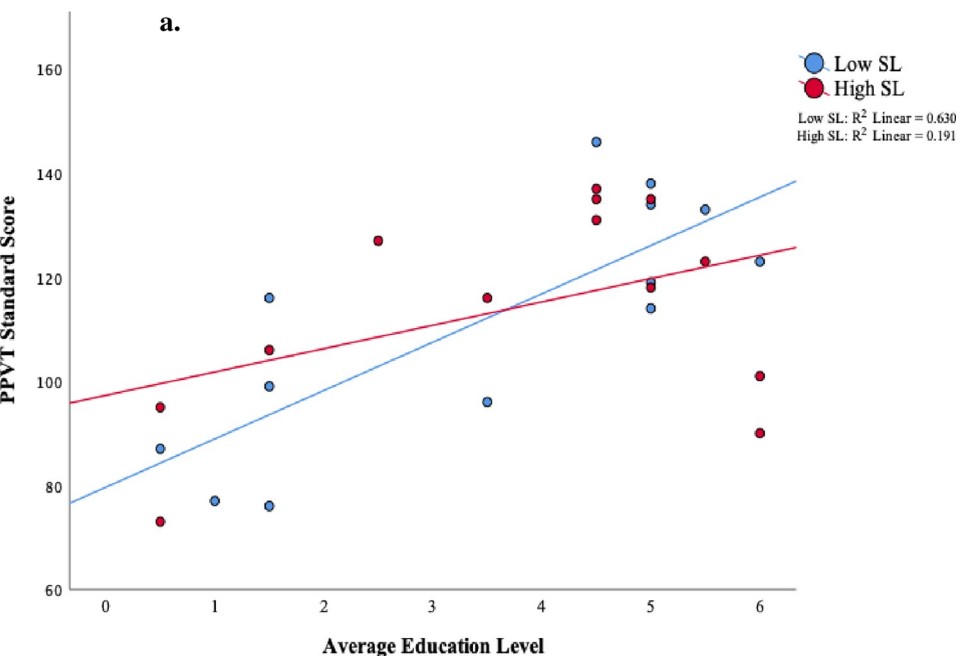

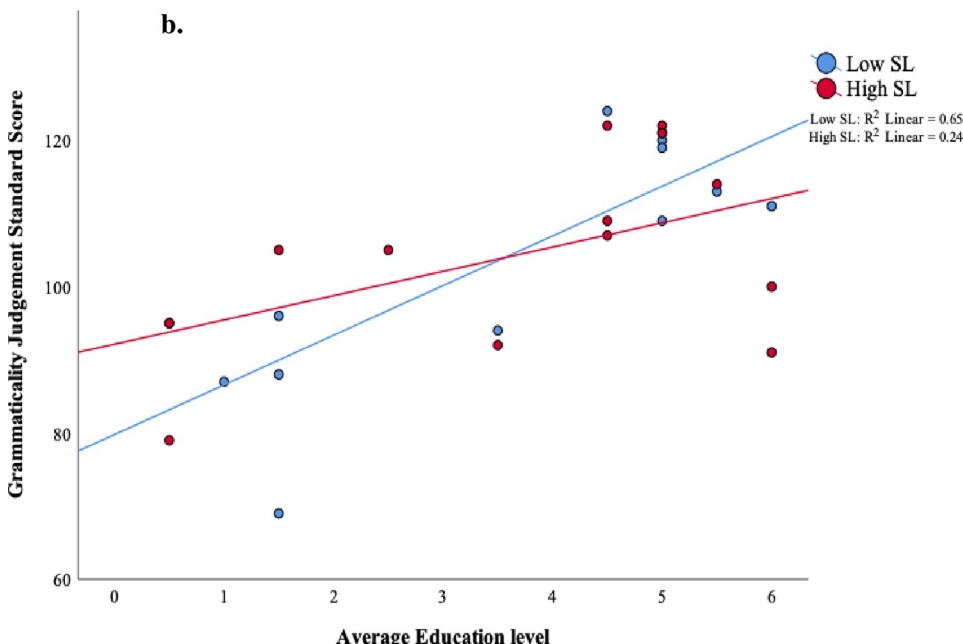

**Fig 6. The significant interaction between parental education and statistical learning performance in the second half of the task.** Scatter plots of (a) PPVT and (b) grammaticality judgement scores vs. parental education for low statistical learning and high statistical learning (as measured by ERP amplitudes) during the second half of the task. For illustration, the statistical learning variable was separated into low and high levels by a median split. Parents' average education level: 0 = Less than High School, 2 = Some college, 4 = Bachelor's degree, 6 = PhD/professional degree.

negative effect of low SES on syntactic and lexical language development appeared to be dampened by high statistical learning performance. Conversely, for children with lower statistical learning scores, their language scores were much more sensitive to the effects of SES. Thus,

children who were raised in less advantaged families showed more typical language development if they had good statistical pattern learning skills. To our knowledge, these results are the first to suggest that statistical learning ability plays a moderating role in the relationship between SES and language development in children.

Although previous studies have found visual statistical learning to predict language ability (e.g., [21]), we did not find a main effect of statistical learning ability on language measures in our sample. Previous work examining the relationship between statistical learning and language has generally been done solely with participants of middle and upper SES levels; it may be that the general relationship between statistical learning and language is different depending on SES. Thus, it is possible that a main effect of statistical learning ability is not apparent because the effect is different for participants of different SES within our sample.

In contrast to the ERP measures, there was no such moderating effect observed using RTs as the measure of statistical learning. On the one hand, the RT and ERP difference score measures were significantly correlated with each other, revealing a coupling between the behavioral indication of statistical learning and the associated neural response. On the other hand, it is likely that these two variables are measuring slightly different aspects of statistical learning. The ERPs are time-locked to the onset of the predictor stimuli before the target appears and therefore seem to reflect the recognition–i.e., a modulation of attention [11, 71]—that certain predictors are cues for the occurrence of the target (i.e., a form of predictive processing). Alternatively, the reaction times are a measure of the behavioral responses following the occurrence of a target and therefore reflect the reaction to the target, and not a prediction that the target will occur. Thus, in the current design, the ERPs likely index perceptual or attentional neural processes associated with making a prediction about the upcoming stimulus as well as response preparation whereas the RT measures reflect an (implicit) motor response (see [57] for a similar perspective). While both measures appear to be related to each other, each one is an index of slightly different aspects of statistical learning, which may be why one but not the other shows a moderating effect of SES on language. In this case, attention-driven predictive processing shows a moderating effect, but the reactive motor response does not. This in turn suggests that predictive processing, such as predicting which syllables, words, or other linguistic units will occur next in speech, may be the crucial link between statistical pattern learning ability and language processing [8].

Although up to this point we have considered statistical learning ability an intrinsic or biological factor and SES an environmental one, it is also possible that children's statistical learning may have been shaped by the environment in which they were raised, similar to other studies showing that low SES is associated with atypical neural development [33, 56]. However, the lack of a significant correlation between SES and statistical learning scores and the non-significant effects of SES on statistical learning in the present study suggests that what drives variation in learning is not due to SES. In fact, what exactly drives variability in implicit and statistical learning remains an open question, which is hindered by some learning tasks exhibiting poor psychometric properties [72, 73]. We believe that the current findings should be considered in light of the fact that there are many different tasks and ways to measure statistical and related forms of learning, and it is not currently clear to what extent these different tasks are or are not related to one another (see [74], for further discussion on this and other challenges related to statistical learning research). Thus, future research should continue to investigate the relationships among different aspects of learning, SES, and language using a variety of tasks and methods.

Finally, we should note that parental education level is just one component of SES. Various components of SES may be individually and/or cumulatively influencing children's linguistic abilities. The predictive validity of parent education has also been shown in other research

demonstrating that it is a reliable predictor of linguistic outcome in children [36, 45, 75]. Future studies might fruitfully focus on using additional variables that are related to the general construct of SES as well as exploring these effects in younger children. Such variables include, but are not limited to, neighborhood context, household structure, socioemotional stressors, and level of cognitive stimulation. It is important to note that we collected parent education level as an ordinal variable, and we averaged both parents/caregivers' education level together to create the parental education level variable. We are aware that as with any ordinal variable, the distance between each category is not known; However, in this case, due to the lack of consensus regarding this issue and ease of interpretation we decided to assume linearity between categories of this SES variable. We should also point out that, in our sample, there is a slightly higher proportion of African American participants in the lower SES group ($N = 7$) compared to the higher SES group ($N = 1$), raising the possibility that cultural differences could also be playing a role in these analyses. Despite the multitude of factors that are likely impinging on individual differences in children's language and statistical learning ability, it is striking that a significant moderating effect was observed with our variables, which suggests that parent education and statistical pattern learning ability together impact language outcomes. So, too, this study was limited by a small sample size ($N = 26$). Future research is needed to replicate and extend these findings using a larger sample.

In sum, this research provides an important examination of the relationship between statistical learning abilities, the socioeconomic environment, and language development in children. The results suggest that having good statistical learning abilities may confer some level of resilience and can help ameliorate the language disadvantages associated with being raised in a lower SES home environment, offering intriguing new ways to think about the relations between learning, language development, and the social/linguistic environment in which a child is raised. Importantly, this result was obtained using a visual non-linguistic measure of learning. This implies a certain level of domain-generality as the statistical learning task was visual and non-linguistic and yet it impacted spoken/auditory language development (for further discussion of domain-generality and modality-specificity, see [11, 76, 77]. One possible implication of these findings is the prospect of designing intervention programs for children of families with low SES by taking statistical pattern learning abilities into account [78]. For example, recent research has demonstrated that it may be possible to improve statistical learning abilities through targeted computerized training (e.g. [79, 80]). Such an approach may be able to facilitate language development in children who are raised in low SES families by minimizing the impact of being raised in a less than optimal social and linguistic home environment. Our results highlight the need to study language development in children by focusing on the interaction between intrinsic (e.g., cognitive) and extrinsic factors (e.g., the social/linguistic environment) for determining the most effective intervention programs for children raised in impoverished environments.

## Acknowledgments

We would like to thank the following individuals for their helpful feedback on an earlier version of this manuscript: Adam Bosen, Angela AuBuchon, Cathy Carotta, Katherine Gordon, Kaylah Lalonde, and Karla McGregor.

## Author Contributions

**Conceptualization:** Leyla Eghbalzad, Christopher M. Conway.

**Data curation:** Leyla Eghbalzad.

**Formal analysis:** Leyla Eghbalzad.

**Funding acquisition:** Christopher M. Conway.

**Investigation:** Joanne A. Deocampo.

**Methodology:** Leyla Eghbalzad, Joanne A. Deocampo, Christopher M. Conway.

**Project administration:** Leyla Eghbalzad, Joanne A. Deocampo, Christopher M. Conway.

**Resources:** Christopher M. Conway.

**Software:** Leyla Eghbalzad.

**Supervision:** Leyla Eghbalzad, Christopher M. Conway.

**Validation:** Leyla Eghbalzad.

**Visualization:** Leyla Eghbalzad.

**Writing – original draft:** Leyla Eghbalzad, Christopher M. Conway.

**Writing – review & editing:** Joanne A. Deocampo, Christopher M. Conway.

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
