## [Decision Letter · Decision Letter 0]

27 Aug 2019

PONE-D-19-17535

How serial pattern learning ability interacts with the socioeconomic environment to shape children’s language development

PLOS ONE

Dear Dr. Conway,

Thank you for submitting your manuscript to PLOS ONE. After careful consideration, we feel that it has merit but does not fully meet PLOS ONE’s publication criteria as it currently stands. Therefore, we invite you to submit a revised version of the manuscript that addresses the points raised during the review process.

My apologies for the length of time required to get the reviews to you. Presumably because it's summer, I was turned down by nearly two dozen reviewers before getting two reviews! 

Because of my interest and expertise in this topic (and challenges in getting reviewers), I acted as a reviewer as well as editor. I am reviewer 3. Some of my comments echo those of other reviewers, however the other reviewers did not have as deep expertise in EEG or statistics and so I have raised some other questions for you. I agree with the other reviewers that the topic is of great and broad interest and so it is important that the analyses are held to the highest standards. 

We would appreciate receiving your revised manuscript by Oct 11 2019 11:59PM. To enhance the reproducibility of your results, we recommend that if applicable you deposit your laboratory protocols in protocols.io, where a protocol can be assigned its own identifier (DOI) such that it can be cited independently in the future. For instructions see: http://journals.plos.org/plosone/s/submission-guidelines#loc-laboratory-protocols

We look forward to receiving your revised manuscript.

Kind regards,

Aaron Jon Newman

Academic Editor

PLOS ONE

Journal Requirements:

Additional Editor Comments (if provided):

Reviewers' comments:

Reviewer's Responses to Questions

**Comments to the Author**

1. Is the manuscript technically sound, and do the data support the conclusions?

Reviewer #1: Yes

Reviewer #2: Partly

Reviewer #3: No

2. Has the statistical analysis been performed appropriately and rigorously? 

Reviewer #1: Yes

Reviewer #2: I Don't Know

Reviewer #3: No

3. Have the authors made all data underlying the findings in their manuscript fully available?

Reviewer #1: No

Reviewer #2: Yes

Reviewer #3: No

4. Is the manuscript presented in an intelligible fashion and written in standard English?

Reviewer #1: Yes

Reviewer #2: Yes

Reviewer #3: Yes

5. Review Comments to the Author

Reviewer #1: What a fantastic and timely paper! I am very excited about the findings presented in this manuscript. The authors make an excellent case for why this research is necessary, and the argument for investigating the interaction SES and serial pattern learning have upon language development is strong. I also greatly appreciate how straight forward and clear the writing is. That being said, I have some questions related to how the EEG data was analyzed that will potentially require major revision on the part of the authors.

Major:

General:

There are reports of a MMN difference for low versus high probability items in statistical learning paradigms/auditory oddball paradigms. Additionally, there are reports of N400/ERAN differences in similar paradigms. Given that other components could be present that could inform us of additional cognitive processes at play during learning, I am curious why the authors have focused their analysis only on the P300 component?

Introduction:

Lines 65-66: “Alternatively, it is possible that learning ability cannot compensate for such environmental limitations.” Do the authors feel it is possible that environmental factors could influence learning ability early on? For example, coming from a low SES home, where you receive poorer quality/quantity input, or poorer nutrition/higher stress/etc., could influence pattern learning?

Can the authors give a more detailed account of what serial pattern learning ability is, including background literature that links this specific type of learning to language development (with specific examples)?

Additionally, can the authors include more concrete examples of how the P300 has been shown to reflect learning in similar paradigms? Specifically, discussing the previous studies (referenced in line 149) that lead the authors to predict faster reaction times and larger ERPs for HP than LP predictors.

Methods:

Measures:

Can the authors please include a breakdown of how many children were in each age group? Additionally, what was the mean age for each group in Table 1 (i.e. did age significantly differ across SES strata)?

EEG:

What was the sampling rate at collection? Was the data re-sampled?

Was the data low-pass/high-pass filtered?

What was the reference electrode during collection? Was the data re-referenced offline?

How were segments of bad data removed? The authors state these were removed manually, but this occurs at the end of their EEG section, so did they remove them after already epoching the data and running ICA?

The segment (epoch) length does not make sense to me. As I understand it, the stimuli was presented for 500 msec followed by a 500 msec black screen. This allows for a 1000 msec window per stimulus. Why would the authors epoch length then be 1500 msec with a 200 msec baseline? This includes the next predictor in the epoch—correct? Even if the authors are only going to investigate the first 500 msec of the epoch, additional noise is being introduced. If my interpretation is correct, I unfortunately would strongly recommend they epoch the data in a window that is more appropriate to their stimulus and time window of interest (-200 to 800 msec or something similar).

When the authors state that data were “replaced with data induced from surrounding sensors using the computational MATLAB software”, does this mean data was interpolated? Was this a spherical interpolation? Some other method?

How were ERPs created? Specifically, was the data low-pass filtered a second time? Was the data baseline corrected? Were single trials averaged together to obtain a stable waveform? Was this done for each condition and each electrode for every subject?

Results:

I was surprised to see the authors analyzed the time window of 400-700 msec, as the P300 is traditionally 250-500 msec. Can this be considered the same effect?

I greatly appreciate the authors inclusion of all data in Figure 4; however, this emphasizes to me that there are additional effects at play. A data driven approach, involving a cluster analysis (or some other variation where all time windows and electrodes are considered), would better allow for the investigation of all these effects. That being said, if the authors can make a stronger argument for a) only looking at the P300, b) only focusing on the 6 electrodes of interest and c) the time window of interest, I can also appreciate analyses that are rooted in solid apriori hypotheses. At this time though, those hypotheses (or why these choices were made) are not as clear as they need to be to not warrant a stronger, more data-driven approach.

Thank you for the Bonferroni corrections and for the inclusion of order (which turned out to be significant).

Correlation analyses: Given that the age range is very broad (7-12) and amplitude of neural oscillations is known to shift during this period of development (impacting ERP amplitude differentially on the basis of age) I feel as though age needs to be included as a covariate in these correlations. I would recommend the authors run partial correlations with age as a covariate, using the raw behavioral scores and ERP data.

Can the authors explain why they elected not to include both RT and ERP in the same regression model?

Lines 343-344: The authors state that “children who showed high levels of learning, SES had a weaker effect 344 on grammar scores compared to children who showed low learning”. Can they provide the simple slopes (or similar) analysis and relevant statistics to support this claim?

Discussion:

The third paragraph of the discussion is very well-written, but I personally feel cuts short the impact of these findings! Can the authors please add a sentence or two that bolsters how stronger serial pattern learning skills may serve as a protective factor for grammar abilities in low SES children? I realize similar statements are included, but would love to see it more strongly stated as I feel this finding is not to be overlooked.

Thank you for your discussion of the null RT results in comparison to the ERP results.

In regards to grammar being significantly influenced by the interaction term, but not vocabulary, I appreciate the authors discussion and agree fully with the reasons they raise. Do they also believe the age of participants could be a factor? Did the groups significantly differ on vocabulary versus grammar? Could it be that ongoing development of complex grammar over this broad developmental time window introduces more variability, as compared to vocabulary (group means/SD would address this I believe)?

Thank you for your acknowledgement of the broad scope of ways SES may be measured.

Minor:

Introduction:

Lines 56-57: Can the authors expand on what they mean by “variations in language exposure” (i.e. conversational turns, quality/quantity of maternal input, etc.)?

Lines 92-96: I feel these lines are better suited for the methods section.

Methods:

Line 101 & 105: Please add the standard deviation of age

How many filler stimuli were present? How did this influence the embedded statistics of the task?

It is not entirely clear in the methods section that children are not tracking the colors, but that they are simply to press when food appears. The figure certainly helps with this, but could the authors explicitly state that the children were not pressing for a target hat, but rather when the food appeared, and that the hats were simply changing colors “in the background”?

Was there a training period?

Results:

Please italicize all M/SD/p/N etc. as per APA

Line 248: Why is RT mean, L-H, but ERP amplitude, H-L?

Line 250: What is the outlier labeling rule?

Line 289: Can you please include the two groups’ descriptive statistics created by the median split analysis? Also include group means/standard deviations on all behavioral measures.

Reviewer #2: Thank you for giving me the opportunity to read this very interesting and well written paper on children’s processing of visual serial information and its relationship to SES. The paper raises two questions that are deeply related to questions around the impact of statistical learning (here referred to as visual serial processing) and language outcomes. The first asks whether statistical learning differences among children might relate to SES differences in language outcomes measured as vocabulary and grammatical learning. The second asks whether visual learning patterns might moderate the effects of SES. Using ERP and reaction time measures to a visual serial test. The authors find that that while there were no main effects of SES on these learning abilities, learning ability plus SES did provide some interesting relationship to language outcomes – here grammar. In particular, children with higher learning scores (measured through the reaction times for learning the probabilities in the statistical learning task) did better on grammar even if they were from low income homes.

The paper does an interesting job of relating aspects of language processing with aspects of language outcome in a statistical learning procedure. Yet, there were significant methodological problems with the manuscript to make me question the full force of the findings.

First, I did not understand why only 38 children were studied. Seems like a small N especially when broken down by SES and learning ability. Also about half of the children were black and I am hoping this is not a confound with SES though the paper does not allow me to tell this.

Second, the age band seems wide – 7 to 12 year olds? Why?

Third, there is other work on efficiency of processing which I would think is quite related to the work discussed here, but it is never cited. (see work by Fernald). There the authors do find the SES is directly related to the processing efficiency.

Fourth, I did not understand a fair number of the decisions on measures. For example, why would you measure learning ability on the same task that you measure visual serial outcome ability? And why do this when there were a host of other cognitive measures that would have allowed these to be treated independently? Unclear. And why only measure this in the second half of the task? To measure real abilities you would want to check out the slope of learning across both halves of the task rather than only when the students have learned the material.

Fifth, on line 85 I am told that past literature focuses on the P300 but the study then shifts by line 208 to 400 to 700. Why?

Sixth, I worried that the statistical treatment was all over the place and largely done to reveal findings rather than to do a comprehensive look at the data. Sometimes ANOVAs are used, sometimes continuous variable assessments like multiple regression, sometimes independent t-tests (line 287) and sometimes visual inspection (line 207). I should think you would want a comprehensive way to explore the data with very clear rationale and precedence for the decisions made.

In short, this is a potentially interesting question that leaves the reader wondering why certain choices were made in sample, methods and analyses and without a better understanding of these variables, it is hard to know how to interpret the findings in a coherent way. As of now, even what is reported seems a tad stretched –statistical learning patterns are related to language outcomes for children who learning them better, but only for language outcomes in vocabulary and only when measured after they have some experience with a task.

Reviewer #3: 1. The introduction is very short and lacks clear hypotheses. Requires a more detailed literature review of the P3, and why it is the component of interest here. Previous work has identified other components in serial tasks such as this, so it’s not clear why the P3 is the focus (especially since the results do not look anything like a P3).

2. Further to that point, and perhaps my biggest concern with this paper: the ERP effect you report looks nothing like a P3 in either the waveform plots or scalp maps. A P3 typically peaks around 300 ms (although there is variance), looks like a single waveform peak, and is maximal over Cz/Pz. Your effect is significantly later in time, looks more like a plateau than a peak, and has a scalp distribution in which the positivity is largest over lateral electrodes. As noted below, it appears you used CZ as a reference so it’s possible that this will look more like a P3 with appropriate re-referencing. But regardless, you should (a) in the Introduction clearly describe the predicted characteristics of the P3, supported by citations (ideally in studies as similar to yours as possible, since task and stimulus parameters will influence how it looks), and (b) in interpretation, provide a rationale for interpreting your effect as a P3, or consider that this may not actually be a P3 and if so, what it might be instead.

3. The hypothesis that “we expected to observe a P300 effect reflecting learning,” (line 85) is insufficient in detail. What is the nature of the predicted P300 effect? Will it be larger or smaller after learning, and how in relation to your experimental conditions? Obviously, this also necessitates describing your experiment in sufficient detail in the introduction that the reader understands what the experimental conditions are.

4. As well, it’s insufficient in the introduction to simply list the behavioural methods that you included. Please be clear on how you expected these to relate to the ERP effects and behavioral learning effects, again in terms of things like larger/smaller P3/RTs to condition X relative to condition Y.

5. The rationale that is provided for the P3 is weak. A “shift in attention to stimuli…” (line 90) doesn’t make sense because in your study, there is no attention shifting – there is a single stimulus on the screen at a time and subjects are expected to be attending to all.

6. The manuscript contains no indication that the study complied with relevant ethical guidelines. In the Participants section please clarify whether participants provided informed consent, whether an ethics board reviewd the study, and whether the practices were in compliance with the Declaration of Helsinki

7. In Methods, please clarify what was used as your SES measure. This becomes clear only later. You do state “We used the average of both caregivers’ education level as a measure of SES.” (line 115) but in saying “a” measure and not “the” measure, it leasves ambiguity.

8. Further to the above point, in the Results section it is misleading to refer to your measure of average parental educational level as SES. Please refer to it as “parental education level”. In the Discussion you can generalize as SES – but please again include the caveat that in this case you used education as a proxy for what is typically described as SES, which includes income and types of employment held by the parents.

9. Learning Task (Methods): please clarify the number of filler trials included

10. A hypothesis appears in the methods on line 150. Hypotheses should be in the Introduction, not buried in the Methods

11. EEG analyses are a problematic area. First off, as noted by another reviewer, you provide insufficient detail on the EEG methods and analysis. You should read and follow published guidelines for ERP studies:

Keil, A., Debener, S., Gratton, G., Junghöfer, M., Kappenman, E., Luck, S., Luu, P., Miller, G., Yee, C. (2013). Committee report: Publication guidelines and recommendations for studies using electroencephalography and magnetoencephalography Psychophysiology 51(1), 1 21. https://dx.doi.org/10.1111/psyp.12147

12. In particular, please (at least) provide the sampling rate used during data collection, ALL preprocessing steps performed, the name and version of software used (appears to be EEGLAB, but “Matlab” is insufficient). Also clarify the method used for electrode interpolation.

13. What was the reference electrode used during recording? Was the data re-referenced during preprocessing? If not, this is a significant problem that likely influenced your results. The EGI system by default uses Cz as the reference during recording. Cz is around where the maximal P3 effect would be observed, so in using this as reference you would effectively subtract the P3 from your data. This indeed is what is suggested by inspection of your scalp topography maps. If you are unaware of issues around re-referencing, you should consult Luck’s book or (shameless self-referencing) my recent book “Research Methods in Cognitive Neuroscience” which addresses this issue in detail, with examples.

14. Please also read and consider Luck, S., Gaspelin, N. (2017). How to get statistically significant effects in any ERP experiment (and why you shouldn't) Psychophysiology 54(1), 146-157. https://dx.doi.org/10.1111/psyp.12639 - your analysis appears to follow many of the “worst practices” highlighted there, notably post hoc selection of electrodes and time windows based on visual inspection. This is particular true of the time window – ayone predicting a P3 on the basis of the literature would focus on an earlier time window than 400-700 ms, whereas this is very obviously the time window one would pick based on post hoc visual inspection of the data.

15. There are some systematic problems with how statistics are reported and interpreted. For one, the treatment of interactions is incorrect. For example, in the behavioral results section you report an interaction of predictor condition and block. However, the follow-up statistics you provide do not fully test the interaction; you report comparisons of condition separately in the first and second blocks, but then only qualitatively compare the magnitude and direction of these differences between blocks. Properly explaining the interaction requires that you statistically compare the condition difference, between blocks. The same is true for your ERP analysis. As it turns out, this was done, but it is reported later in section 3.3. This section should be removed because what is contained there are really the analyses you should be presenting to interpret your interactions.

16. Incidentally, section 3.3 (line 247) has inconsistent heading formatting -it is the only section with a numbered heading and is nested under “neurophysiological evidence of learning” even though it does not exclusively relate to ERP evidence).

17. Lines 223-235 present a qualitative description of scalp distributions of the effects. This includes evaluative language like “the voltage response in posterior regions is stronger”. This is not appropriate. If you wish to make such claims you need to support them quantitatively with statistical analyses.

18. Please indicate in the legends for Figs 5-6 what the error bars represent (95% CIs? SE?)

19. Figure legends are poorly formatted. They are inline with the body text of the manuscript which makes reading the text hard. As well, since the figures appear at the end, it is very hard to interpret the figure due to the need to scroll back and forth between legends and figures. Please provide legends immediately before figures as is standard practice.

20. Please provide a rationale for describing your RT and ERP results as reflecting learning “ability”. It seems that you have a single measure of learning and are generalizing from this to a more generalized “ability”.

21. Correlation analyses: You treat SES as a continuous variable in the correlation analyses. I’m not convinced this is appropriate. The measure is not continuous, it’s ordinal. Furthermore, contra the assumption of a linear measure, the conceptual “distance” between the levels of the SES variable are not necessarily even steps. For instance, is the difference in SES for parents with a PhD relative to those with a professional degree really the same as between high school and some college? See, for example, https://www.frontiersin.org/articles/10.3389/fams.2017.00015/full for a description of the issue. This is an area in which there is a lack of consensus, and in many cases the linearity assumption is approximately correct. However, you need to explicitly recognize and address this issue in your manuscript, and justify treating SESas continuous. See, e.g., https://www3.nd.edu/~rwilliam/stats3/OrdinalIndependent.pdf.

22. Please clarify whether multiple comparison correction was used in computing the significance of the 45 correlation values reported in Table 2

23. The “influence of SES on learning” section applied a median split to the SES data. First off, please clarify what cutoff this resulted in – from Table 1 my guess is between 3 and 4 on your scale, but this should be made explicit. Assuming this is so, an important thing to address in your Discussion is that this analysis differentiates between people with college degrees (bachelor’s or higher) and those without. As noted above, this is rather different from a more general construct of “SES”

24. Discussion: Around line 404, consider the alternative explanation that in addition to “making a prediction about the upcoming stimulus”, your ERP effects reflect response preparation.

25. Line 448: You state, “…the lack of a significant correlation between SES and learning ability and the non-significant effects of SES on learning in the present study suggests that what drives variation in learning is not due to social environmental factors such as SES”. You are overstepping your data here, in the last part of the sentence. Please remove “social environmental factors such as” and just say SES, as you have no evidence on other social-environmental factors to justify this claim.

26. Lines 464-465: please quantify the proportion of subjects in each SES group by race (either here or earlier when the SES distribution is reported)

27. Lines 473-476: You suggest that serial pattern learning abilities might help “ameliorate language disadvantages”, but elsewhere my interpretation of your writing was that you thought that relationships between grammatical and serial learning abilities are because they rely, at least in part, on the same neurocognitive machinery. If that is so, then why do you imply a unidirectional influence of serial learning abilities on grammar? Would grammar training not be equally predicted to boost serial learning abilities?

6. PLOS authors have the option to publish the peer review history of their article (what does this mean?). If published, this will include your full peer review and any attached files.

Reviewer #1: Yes: Julie M. Schneider

Reviewer #2: Yes: Kathy Hirsh-Pasek

Reviewer #3: Yes: aaron newman

---

## [Author Response · Author response to Decision Letter 0]

4 Sep 2020

Our detailed response to reviewer/editor comments has been uploaded with this submission.

---

## [Decision Letter · Decision Letter 1]

19 Oct 2020

PONE-D-19-17535R1

How statistical learning interacts with the socioeconomic environment to shape children’s language development

PLOS ONE

Dear Dr. Conway,

Thank you for submitting your manuscript to PLOS ONE. One of the original external reviewers has re-reviewed the manuscript, as have I. After careful consideration, we feel that the manuscript is acceptable for publication, pending minor changes as detailed below. As some of those requested changes reflect errors in reporting statistical analysis, the manuscript does not fully meet PLOS ONE’s publication criteria as it currently stands.

Therefore, we invite you to submit a revised version of the manuscript that addresses the points raised during the review process.

We look forward to receiving your revised manuscript.

Kind regards,

Aaron Jon Newman

Academic Editor

PLOS ONE

Reviewers' comments:

Reviewer's Responses to Questions

**Comments to the Author**

1. If the authors have adequately addressed your comments raised in a previous round of review and you feel that this manuscript is now acceptable for publication, you may indicate that here to bypass the “Comments to the Author” section, enter your conflict of interest statement in the “Confidential to Editor” section, and submit your "Accept" recommendation.

Reviewer #1: All comments have been addressed

Reviewer #3: (No Response)

2. Is the manuscript technically sound, and do the data support the conclusions?

Reviewer #1: Yes

Reviewer #3: Partly

3. Has the statistical analysis been performed appropriately and rigorously? 

Reviewer #1: Yes

Reviewer #3: No

4. Have the authors made all data underlying the findings in their manuscript fully available?

Reviewer #1: (No Response)

Reviewer #3: No

5. Is the manuscript presented in an intelligible fashion and written in standard English?

Reviewer #1: Yes

Reviewer #3: Yes

6. Review Comments to the Author

Reviewer #1: The authors of “How statistical learning interacts with the socioeconomic environment to shape children’s language development” have done an incredible job addressing each of my concerns with the current manuscript. More specifically, I greatly appreciate the substantial effort it took to re-epoch and analyze all of the EEG data. I also believe that their decision to proceed with the investigation of the P300 based on apriori predictions to be valid, despite it not being a more data-driven approach. Taken together, I believe the manuscript is greatly improved due to their efforts and will make an important contribution to the field. I recommend it for publication without reservation.

Reviewer #3: # Conway PONE

1) P. 7 Please change the word “impairment in the sentence, “low SES may be associated with impairments to a variety of cognitive abilities including working memory, cognitive control, and language”. The word “impairment” is both loaded, and inaccurate. The citation used to support this statement does not contain the word stem “impair” anywhere in it, and indeed talks of the effects of SES in terms of correlations and gradients, and notes that “…the statistical effects of SES operate across the whole SES continuum…” with little evidence of “threshold” effects. Please use more graded language, such as “low SES may be associated with lower scores on a range of measures of cognitive abilities…”

## Hypotheses:

2) - Hypotheses are improved but still need work. There is no mention of ERPs with respect to the two research questions of the study. That is, there is no prediction as to how SES will relate to ERPs. If SES-related questions are the primary research questions of the study, and no predictions about EPRs were made, this begs the question of why ERP data was collected or presented. Please state predictions regarding ERP data with respect to SES.

## Methods

3) Data from 7 year olds was collected as part of this study, and was included in the first draft. For this second draft, the authors have chosen to remove these data due to low trials/noisiness. While this rationale is acceptable, removing all mention of these participants, and why their data was removed from the analysis, is not. This is not open and transparent scientific practice. Indeed, the statement on p. 7 “We recruited 27 typically developing, monolingual English-speaking children aged 8-12 “ is entirely false, given that the previous version of the manuscript stated, “We recruited 42 typically developing, monolingual English-speaking children aged 7-12…” Moreover, the following sentences in this section provide a rationale that misleadingly implies that the 8-12 year old range was planned a priori (“We chose the age range 8-12 years for four reasons…”). Please include mention of the 7 year olds in the methods and provide your rationale there for removing them from analysis.

### SL Task

4) I was confused by the statement, “Each experimental condition (HP and LP) contained 60 trials.”. It would be helpful to the reader if you moved the paragraph starting with the words, “Each stimulus was presented for 500 milliseconds…” on p. 11 up to appear after the sentence, “The participants viewed a stream of flashing stimuli consisting of hats of different colors presented one at a time with a black background.” On p. 10, or somewhere around there.

### EEG preprocessing

5) The authors report 1 Hz high-pass filtering of there EEG data. This seems excessive for EEG data, especially in light of evidence that even 0.3 Hz filtering can attenuate real ERP components and induce artifactual ones (Tanner, D., Morgan‐Short, K., & Luck, S. J. (2015). How inappropriate high‐pass filters can produce artifactual effects and incorrect conclusions in ERP studies of language and cognition. Psychophysiology, 52(8), 997-1009.). Is this a typo, or can a rationale for this extreme filtering be provided?

6) p. 13, line 7: don’t capitalize “all” after the semicolon

7) p. 13, para 2, line 2 - require period after Jost citation.

8) p. 13, again after a Jost citation a period (and space) are missing, in the sentence ending “…also an a priori decision based on the findings from Jost et al. [56]”

## Results

9) The reporting of behavioural (RT) results is confusing and at times inaccurate.

(a) The first paragraph of the results section mentioned a condition x block interaction, but then provides follow-up t-tests that don’t really support or explain the interaction: the fact that RTs were significantly faster for HP and LP in both blocks is not evidence for an interaction, it’s evidence for a main effect of condition.

(b) The next paragraph refers to “the” main effect of block, as if this had been previously reported (and providing no statistical information to support the claim of significance).

(c) Further, the t-tests used to address whether there was an ME of block aren’t appropriate for that question: they report the differences between blocks separately for HP and LP conditions. By definition, a main effect of block is _collapsed_ across levels of any other condition. The appropriate way to report this is to report the F and p values from the ANOVA for the ME of block. The authors state that this was not significant, so upon providing the F and p values to back up that claim, no subsequent t-tests are necessary.

(d) Further, the main effects of the ANOVA (for both block, and condition; the latter is currently missing) should be reported first, and the interaction second. Note that t-tests appropriate to explaining the interaction are provided, in the latter part of paragraph 2. These need to follow after the reporting of the interaction in the ANOVA

10) Figure 3’s caption needs further detail. The bars are surely indexing the _difference_ in RT between HP and LP conditions, in each half of the experiment, but this is not stated in the caption. Please clarify this fact in the caption.

11) Similar errors in ANOVA reporting are present in the ERP analysis, as in the RT analysis. Specifically, the third line from the bottom of p. 17 starts, “Additional paired sample t-tests exploring the main effect of block…” and then reports t-tests computed separately for the HP and LP conditions. Again, an analysis of block that is broken down by condition is not a test of the main effect of block, it is another way of examining the block X condition interaction. This section goes on to conclude, on the basis of these t-tests, “These results suggest that there is no main effect of predictor condition and block…” - which is again not the way to assess the ME of either condition or block. Please simply report the statistical values (F and p) for the main effects of each factor, that were provided by your ANOVA results.

12) the motivation for the median-split comparisons of high vs. Low SES groups on ERPs and RTs is not clear. The lack of a relationship between these variables was already established by the correlation results in the previous section. Please provide a rational or remove these effectively redundant analyses.

13) The statement in the first line of the last paragraph of p. 21 is conducing and misleading. It states, “…results of the regression analysis using the neurophysiological measure showed that statistical learning ability was a significant moderator of the relationship between parental education and both language measures”. My issue is that the authors are defining a difference in ERP amplitude between the HP and LP conditions as “statistical learning ability”. I’m not convinced that an ERP difference reflects an “ability”, especially in the absence of a parallel significant moderating effect of the behavioral measure of statistical learning. I suggest that the authors use a more neutral and empirically-based term here (e.g., “difference in ERP amplitude”) and provide a interpretation/rationalization in the discussion as to why they equate differences in the ERP amplitude with differences in learning ability.

14) p. 24 line 1, please remove the space after “however” before the comma

15) p. 24 line 3, “As an additional control…” should be the start of a new paragraph

7. PLOS authors have the option to publish the peer review history of their article (what does this mean?). If published, this will include your full peer review and any attached files.

Reviewer #1: **Yes: **Julie M. Schneider

Reviewer #3: **Yes: **Aaron Newman

---

## [Author Response · Author response to Decision Letter 1]

4 Dec 2020

Our response to reviewer/editor comments are in the uploaded file.

---

## [Editor Report · Decision Letter 2]

21 Dec 2020

How statistical learning interacts with the socioeconomic environment to shape children’s language development

PONE-D-19-17535R2

Dear Dr. Conway,

We’re pleased to inform you that your manuscript has been judged scientifically suitable for publication and will be formally accepted for publication once it meets all outstanding technical requirements.

Kind regards,

Aaron Jon Newman

Academic Editor

PLOS ONE
---

## [Editor Report · Acceptance letter]

11 Jan 2021

PONE-D-19-17535R2 

How statistical learning interacts with the socioeconomic environment to shape children’s language development 

Dear Dr. Conway:

I'm pleased to inform you that your manuscript has been deemed suitable for publication in PLOS ONE. Congratulations! Your manuscript is now with our production department. 

Kind regards, 

on behalf of

Dr. Aaron Jon Newman 

Academic Editor

PLOS ONE